# Adenomyosis and Its Possible Malignancy: A Review of the Literature

**DOI:** 10.3390/diagnostics13111883

**Published:** 2023-05-28

**Authors:** Liviu Moraru, Melinda-Ildiko Mitranovici, Diana Maria Chiorean, Raluca Moraru, Laura Caravia, Andreea Taisia Tiron, Ovidiu Simion Cotoi

**Affiliations:** 1Department of Anatomy, “George Emil Palade” University of Medicine, Pharmacy, Sciences and Technology, 540142 Targu Mures, Romania; liviu.moraru@umfst.ro; 2Department of Obstetrics and Gynecology, Emergency County Hospital Hunedoara, 14 Victoriei Street, 331057 Hunedoara, Romania; 3Department of Pathology, County Clinical Hospital of Targu Mures, 540072 Targu Mures, Romania; chioreandianamaria@yahoo.com (D.M.C.); ovidiu.cotoi@umfst.ro (O.S.C.); 4Faculty of Medicine, “George Emil Palade” University of Medicine, Pharmacy, Sciences and Technology, 540142 Targu Mures, Romania; raluca.moraru@umfst.ro; 5Division of Cellular and Molecular Biology and Histology, Department of Morphological Sciences, ”Carol Davila” University of Medicine and Pharmacy, 050474 Bucharest, Romania; laura.caravia@umfcd.ro; 6Faculty of Medicine, “Carol Davila” University of Medicine and Pharmacy, 050474 Bucharest, Romania; taisia_andreea@yahoo.com; 7Department of Pathophysiology, ”George Emil Palade” University of Medicine, Pharmacy, Science, and Technology of Targu Mures, 38 Gheorghe Marinescu Street, 540142 Targu Mures, Romania

**Keywords:** adenomyosis, cancer, endometriosis, immunohistochemistry, tumour markers, malignant transformation

## Abstract

Cancer arising from adenomyosis is very rare, with transformation occurring in only 1% of cases and in older individuals. Adenomyosis, endometriosis and cancers may share a common pathogenic mechanism that includes hormonal factors, genetic predisposition, growth factors, inflammation, immune system dysregulation, environmental factors and oxidative stress. Endometriosis and adenomyosis both exhibit malignant behaviour. The most common risk factor for malignant transformation is prolonged exposure to oestrogens. The golden standard for diagnosis is histopathology. Colman and Rosenthal emphasised the most important characteristics in adenomyosis-associated cancer. Kumar and Anderson emphasised the importance of demonstrating a transition between benign and malignant endometrial glands in cancer arising from adenomyosis. As it is very rare, it is difficult to standardize treatment. In this manuscript, we try to emphasize some aspects regarding the management strategy, as well as how heterogenous the studies from the literature are in terms of prognosis in both cancers that develop from adenomyosis or those that are only associated with adenomyosis. The pathogenic mechanisms of transformation remain unclear. As these types of cancer are so rare, there is no standardised treatment. A novel target in the diagnosis and treatment of gynaecological malignancies associated with adenomyosis is also being studied for the development of new therapeutic concepts.

## 1. Introduction

Adenomyosis is a common, benign gynaecological disease characterised by the extension of endometrial tissue into the myometrium [1]. It affects mostly women of late reproductive age, including women in menopause [2,3]. Pelvic endometriosis is a common disease affecting 7–15% of women of reproductive age [4,5], and it is expected to disappear with advancing age as it is oestrogen-dependent [6]. However, it has a prevalence of 2–4% in postmenopausal women, and in these cases, it can be associated with adenomyosis with a rare possibility of malignancy [6].

Adenomyosis is characterised by the presence of aberrant endometrial tissue outside of the uterine cavity in intra- and extra-abdominal sites [4,5,7]. It can also occur in the cervix, round ligament, abdominal scars [4,7], pararectal space, paraovarian region, parametrium, liver, appendix and mesentery [8]. It is often associated with endometriosis [7].

Cancer arising from adenomyosis is very rare, with transformation occurring in only 1% of cases [4,5,9] and in older individuals [10]. The first case of clear cell and endometrioid carcinoma arising from adenomyosis was described in 1897 [3]. Adenomyosis is more commonly associated with endometrioid carcinoma, but clear cell carcinoma is also observed [3,4]. Certain cell types are involved, such as epithelial and mullerian types, with sarcomas described as well [11,12].

Even if endometriosis, adenomyosis and cancers have common manifestations, the pathogenic mechanism of malignant transformation remains unknown. A better knowledge of pathogenesis aids in diagnostic and therapeutic management [11,12]. The histological description is important for standardising the description of cancers developed from adenomyosis and to differentiate them from those that appear simultaneously with adenomyosis without knowing the exact relationship between them [13,14]. These concomitants are important because they modify the therapeutic strategy [14].

As cancer arising from adenomyosis is very rare, adenomyosis, endometriosis and cancers may share a common pathogenic pathway, though the pathogenic mechanism still remains to be established. There is also a need to have a standardised treatment.

## 2. Materials and Methods

We used the PubMed and Cochrane databases to select relevant articles for this descriptive review, using the keywords adenomyosis, cancer and endometriosis.

Furthermore, we examined the reference lists of highly cited articles for an adequate selection of studies that were considered suitable. The articles were written in English and published between 2012 and 2023. The inclusion criteria were based on (a) imaging and biomarkers, (b) anatomy, (c) pathogenesis, (d) symptoms, (e) histopathological examination, (f) differential diagnosis, (g) metastases, (h) management and (i) prognosis. Other studies which were written in a language other than English and were published before 2012 were excluded from our data analysis. Two authors then manually and independently screened relevant articles according to the inclusion criteria, which were based on PRISMA 2020.

The search of the literature search identified 768 potentially qualified articles. Duplicates were excluded after the initial search. The others were excluded after full-text screenings based on our exclusion criteria mentioned above or in the cases of poorly designed studies or missing data.

### PubMed Search Strategy

The literature search identified 768 potentially qualified articles. Duplicates were excluded after the initial search. The others were excluded after full-text screenings based on the exclusion criteria mentioned above or in the cases of poorly designed studies.

Utilizing the PubMed Advanced Search Builder, our investigation was founded on the subsequent aspects:

a. Primary Concepts:Adenomyosis;Cancer (including Neoplasms);Endometriosis.

b. Publication Date Range:From 2012 to 2023.

c. Key Aspects/Fields of Study:Anatomy and histology;Etiology or pathogenesis;Management or organisation and administration;Symptoms;Imaging;Biomarkers;Histopathologies or pathologyManagement.

## 3. Pathogenesis

The pathogenesis of a malignant transformation in adenomyosis appears to involve inflammation and elevated levels of IL1 and IL6 [4]. The underlying mechanisms may involve genetic mutations, epigenetic changes, and tumour suppressor gene alterations in adenomyosis [11]. IL-37 is also involved in adenomyosis. IL-37 was discovered through a bioinformatics analysis in 2000 and is a member of the IL-1 family. Oestrogen and progesterone do not have an effect on the IL-37 protein in cancer cells. While IL-37 does not affect the proliferation and colonisation of cancer cells, it suppresses the migration and invasion ability of endometrial cancer cells. Furthermore, a decreased expression of MMP2 via the Rac/NF-kB signalling pathway in cancers is also observed [12].

Adenomyosis, endometriosis and cancers may share a common pathogenic mechanism that includes hormonal factors, genetic predisposition, growth factors, inflammation, immune system dysregulation, environmental factors and oxidative stress [11] (Table 1). Endometriosis and adenomyosis both exhibit malignant behaviour [4,5,11,12].

The pathogenic mechanisms of transformation still remain unclear [11].

Annexin ANXA2, a protein that is increased in angiogenesis, metastasis, endometrial growth and the epithelial–mesenchymal transition, is implicated in pathogenesis. Similarly, invasion, metastasis and tissue growth occur in oxidative stress and inflammation. A KRAS mutation in the V-Ki-ras2 Kirsten rat sarcoma viral oncogene homolog is responsible for increased growth factors and hinders the tumour response to progesterone, interfering with treatment [10,11,12,13,14].

This condition may be associated with genetic mutations at the CTNNB1-encoding B-catenin (cadherin-associated protein) level [12]. The ARID1A (A-rich interactive domain-containing protein 1A) is a tumour suppressor gene that is disrupted and interacts with p53 [12,14,15]. It is associated with heterozygosity loss in adenosarcomas, and other mutations such as JAZF1-SuZ12, EPC1-PHF1 or PTEN loss have also been reported [15,16]. Reactive oxygen species (ROS) can cause DNA damage, and various inflammatory factors, including cyclooxygenase 2 (COX-2), TNF-alpha, toll-like receptors (TLRs), nuclear factor kappa B (NF-κB) and macrophages, as well as IL-6 and IL-10, are implicated in pathogenesis. Ceasing and reversing the epithelial–mesenchymal transformation may function as a therapeutic strategy [10,11].

Tamoxifen may be implicated in the malignancy of adenomyosis, which is a risk factor for the development of genital cancers but has a good prognosis [6,10,11,12]. While direct malignancy occurs in less than 1% of cases and is associated with a poor prognosis, the potential for transformation is poorly understood, even though it shares pathogenic pathways. Further studies are needed [11].

Risk factors include pelvic irradiation, and the laparoscopic morcellation of benign uterine tumours can lead to unexpected malignancies [9,15,17]. However, endobag morcellation can be used with minimal risks. Other risk factors associated with endometrial cancer include BMI, hypertension, hyperinsulinemia and prolonged exposure to oestrogens [3]. Hormone replacement therapy is the most common risk factor, although ectopic endometrial tissue can produce oestrogens through autocrine and paracrine effects, which may explain the persistence and recurrence of tumours in menopause [6].

Metalloproteinases 2 and 9 appear to be involved in tumour invasion, while PCNA is a marker of proliferation [18]. They can be used in an immunoreactive score which has been previously used in colorectal endometriosis and endometriotic invasion [18]. However, MMP2 and MMP9 increase during a malignant transformation, so they can be used as indicators of malignancy [18].

Regarding the theories of adenomyosis, there are two main hypotheses: the first is that tissue injury and repair in the endometrium lead to stromal invagination into the layer of myometrium, which is associated with environmental factors; the second is based on the metaplasia of displaced embryonic pluripotent Mullerian remnants or stem cells. An essential involvement in adenomyosis is the epithelial–mesenchymal transition, as previously mentioned [16].

## 4. Anatomy

The uterus (Figure 1) is a cavity organ with three layers. There is the endometrium, which undergoes cyclic changes through the basal and functional layers. It is separated from the myometrium by the junctional zone and is covered by the serosa. The junctional layer is affected when the endometrium invades the myometrium and forms adenomyosis. Therefore, in cases of malignant transformation, staging is not as clearly defined as in endometrial cancer, as the ectopic endometrium is present deep within the myometrium, closer to the lymphovascular space. Metastasis can occur through both tubal reflux and invasion into the lymphovascular space and sometimes into the peritoneal cavity with the formation of ascites through migration along large lymphatic vessels towards the diaphragm [9].

## 5. Diagnosis

### 5.1. Symptoms

Symptoms of adenomyosis include irregular bleeding, painful menstruation, infertility, dysmenorrhea and anaemia [1,4,5,9,19,20]. In cases of hepatic metastasis, hypochondriac pain may occur [21]. Adenomyosis can also be associated with diabetes mellitus, hypertension and hydatidiform mole [19]. However, it can also be asymptomatic at times [3].

### 5.2. Imaging and Biomarkers

Adenomyosis is visible on CT and MRI as a cystic transformation of a leiomyoma, with a heterogeneous appearance and sometimes multilocular, exophytic lesions [4,5,7,22]. An increase in the tumour marker CA 125 is observed [1,4,5,7,20,22], and other biomarkers that may increase include CA 19-9 [23]. The MRI appearance has been described as a “fish in the net”, with the myometrium resembling “Swiss cheese” in a malignant transformation [20]. In cases of a malignant transformation of adenomyosis, only a darker junctional zone may be observed [23]. CT scans of the abdomen, pelvis and chest may be performed to detect metastases, and ascites may sometimes be observed [5,23]. Large pleural effusions may be detected in the chest, requiring thoracentesis [9]. PET-CT scans may show distant metastases [7]. Transvaginal ultrasound can identify a tumour-like formation that is similar to a fibroid but with cystic degeneration [4,9], and ground-glass echoes may be observed [5,11]. Ultrasound for adenomyosis follows MUSA guidelines [11]. PET-CT scans may also show distant metastases [7].

Hysterosalpingography may reveal adenomyosis [24], and it may be discovered incidentally on pathological anatomy after surgery [24]. Hysteroscopy with curettage may help diagnose adenomyosis in some situations [19], but it is not helpful in sarcomas as neither are biomarkers in the case of sarcoma [17]. Colonoscopy proves useful in detecting metastases at the intestinal level [15].

### 5.3. Histopathological Examination

Histopathological examination is the golden standard for diagnosis and is used to confirm clear cell adenocarcinoma arising from an adenomyotic cyst. Immunohistochemical examination reveals an inverse relationship between the tumour and the endometrium: if the endometrium is positive for an oestrogen receptor and negative for p53, the tumour is negative, and vice versa [4]. Hysteroscopy and curettage usually show a normal endometrium [5].

Macroscopically, the tumour can present as a polypoid mass, which is visible on transvaginal ultrasound, in the endometrium of a postmenopausal woman [7].

Sampson’s or Colman’s criteria for the adenocarcinoma–adenomyosis association include: carcinoma not located in the endometrium or elsewhere in the pelvis, carcinoma arising from the epithelium of adenomyosis, the absence of invasion from another source, the presence of endometrial stromal cells to support a diagnosis of adenomyosis, a smooth, round contour of the surrounding myometrium and adenomyotic glands and endometrial-type stroma within the carcinoma foci [4].

Colman and Rosenthal emphasize that there are some signs of adenomyosis-associated cancer, such as the absence of carcinoma in the surrounding endometrium or carcinoma arising from the epithelium of adenomyosis, and endometrial stromal cells must surround the aberrant glands to support the point of origin as adenomyosis [3,22]. Adenomyosis may have a protective effect against cancer progression; however, despite this, it has a poor survival rate [3].

Kumar and Anderson emphasised the importance of demonstrating a transition between benign adenomyotic endometrial glands and carcinomatous glands to confirm a diagnosis of ectopic, endometrium-derived adenocarcinoma [4]. The diagnosis of a tumour derived from adenomyosis requires characteristic endometrial stromal surroundings of the epithelial glands [5]. Both epithelial and mesenchymal histologic elements are involved in this highly malignant tumour, which is associated with a worse prognosis and higher rates of metastatic disease [25].

Histologically, glandular cells without atypia can be seen, in addition to sarcomatous components demonstrating pleomorphism and high mitotic rates [7].

The Mullerian tumour, first described in 1974, contains a mixture of benign elements, occasionally mildly atypical glandular components and malignant, unusually low-grade stromal components, smooth muscle metaplasia or subcoelomic mesenchymal metaplasia [1,7,26].

Mullerian adenosarcoma typically arises from ectopic endometrium in the uterine corpus and presents as a polypoid mass in menopausal women [7]. Residual cells from embryogenesis are widely spread throughout the peritoneal cavity, so it can appear anywhere [1,12]. However, it can also arise from the ectopic foci of endometriosis in the cervix, vagina, broad and round ligament, ovaries, extragonadal sites, adenomyosis, salpinx, appendix, urinary bladder, peritoneum, abdominal wall and lymph nodes [12]. It may arise from stem cells originating from the bone marrow or endothelial progenitors, which differentiate into endometrium-like tissue [16]. The roles of genetics, epigenetic factors, inflammation, immune dysfunction and neovascularisation are essential in the development of Mullerian and non-Mullerian regenerating stem cells with endometrial differentiation [14]. FIGO staging is not available as it is extremely rare [7]. Dilated glandular elements and hypercellular stromal elements with benign glandular elements are present. Immunohistochemistry can show positive receptors for the progesterone receptor, oestrogen and CD10 and increased levels of Ki67 [1,7,27].

Endometrial cancer may coexist with adenomyoma, but it is not the same as cancer arising from adenomyoma [3,28].

Immunohistochemistry reveals the expression of the stem-cell-related markers CD44 and CD133, which are commonly found in lymphatic tumours and leukaemias. However, in this case, they appear to be involved in carcinogenesis [29]. CD10 is the most specific marker [1,7,27]. In addition, immunohistochemistry has shown positivity for other less-specific markers such as CK7 (cytokeratin), p53, CA125, Pax8 and Ki67 with a high mitotic index, as well as positive progesterone and oestrogen receptors [3,21]. Recently, Wilms’ tumour 1 (WT1) has emerged as a histological tumour marker [30].

Anatomopathological types include:Epithelial carcinomas;Other Mullerian type tumours, including Mullerian-type mucinous borderline tumours and serous borderline tumours;Sarcomas: adenosarcoma and endometrial stromal sarcoma [12].

Mullerian mucinous borderline tumours commonly occur in the intestine, are uncommon and develop from a ciliated columnar epithelium reminiscent of the fallopian tube [12].

The most common gynaecological cancer in developed countries is that of the endometrium [31]. Both components, epithelial and stromal, can become malignant [17]. The most common is adenocarcinoma, followed by sarcoma, and the frequency increases with age [19]. Endometriosis and adenomyosis share characteristics with malignant tissue invasion.

Adenosarcoma is also uncommon, and it is a mixture of benign and malignant lesions, with the latter comprising low-grade stromal components. It often appears in the endocervix and has risk factors such as pelvic irradiation, hyperoestrogenism and tamoxifen. It frequently occurs in the lesions of adenomyosis [12].

Endometrial stromal sarcoma is a rare, malignant transformation of endometriosis which occurs particularly in endometriotic nodules but also in adenomyosis from pluripotent Mullerian cells. It is sensitive to oestrogens but also has a genetic component [12,32]. Endometrial stromal sarcomas are composed of endometrial stromal cells. They are low-grade sarcomas and account for 10–15 %of pure mesenchymal tumours [17]. The most-studied cancer that develops from endometriosis is ovarian cancer, with the most common being endometrioid and clear cell cancers [5,26,33]. Regarding the cancers that develop from adenomyosis, we can exemplify those found in the uterus (endometrial endometrioid cancer and uterine sarcomas) [10] or in the cervix (adenocarcinoma, squamous cell carcinoma) [34], and last but not least, extrauterine cancers (sarcomas: inguinal, appendix, the round ligament, small bowel or mesentery) [8].

## 6. Metastases

Metastases commonly occur in the omentum, sometimes in the ovaries, and ascites may be associated [1,9]. Hepatic metastases with diaphragmatic involvement have been reported [21]. Metastasis may occur following tumour morcellation in laparoscopic surgery for benign tumours, resulting in metastases in the small intestine and liver parenchyma. This highlights the necessity for counselling and shared decision-making prior to morcellation procedures [15].

## 7. Associations with Other Types of Cancer

Other types of cancer can appear in association with adenomyosis without our awareness of any connection or predisposition in this regard. These types of cancers include lymphoma, melanoma, ovarian, endometrial or breast cancers. Endometrial cancer can coexist with adenomyosis without having developed from it [3,28]. Synchronous cancers have become frequent recently, such as endometrioid adenocarcinoma of the ovary and clear cell carcinoma, and their coexistence modifies the therapeutical approach [6,12,23].

## 8. Differential Diagnosis

Differential diagnosis can be difficult, such as with various types of uterine tumours, including adenofibromas, adenosarcoma and carcinosarcoma, which consist of a mixture of epithelial and stromal components. The aetiology and prevalence of this transformation are unknown. Gene-encoding proteins with antioxidant properties may influence susceptibility to malignant transformation. The association of the apolipoprotein E2 allele may also be involved [5,35].

## 9. Management

The management of these cancers involves surgical tumour excision with free margins and adjuvant radiotherapy and chemotherapy [5]. In some cases, radiotherapy is performed prior to surgery to reduce tumour extension and has a survival advantage [1,5,7,25]. The tumour block may present necrosis and haemorrhage, and the capsule may rupture [5].

Tumour cytoreduction via the removal of the uterus, omentum, lymph nodes and affected areas is the optimal intervention [1,9,22]. However, radical surgery with lymphadenectomy remains in debate, with some opting for fertility preservation surgery that is associated with adjuvant chemotherapy and hormone therapy [1]. Other studies suggest that lymphadenectomy can be avoided in cases of low-risk patterns. For this purpose, we use the Mayo triage algorithm. Lymphadenectomy is useful in cases of individuals with high-risk factors [3]. These risk factors have been established as an age >60 years, grade 3 LVSI, deep myometrial invasion and serous or clear cell histology [3].

In some cases, the histological diagnosis is a surprise, and surgery is performed for another pathology. In this situation, re-intervention is sometimes necessary for completion with a bilateral salpingo-oophorectomy and lymphadenectomy [3].

As these cancers are very rare, it is difficult to standardize treatment. Adjuvant platinum-based chemotherapy is necessary to prevent recurrence [5,7,25]. Overall survival is low, and the prognosis is poor [7,25]. Recently, a better response to chemotherapy with paclitaxel and carboplatin has been observed [21,23].

Hormone therapy can be added: progesterone, letrozole and tamoxifen are used, but not routinely [1,9]. Tamoxifen is even included among the risk factors for the malignant transformation of adenomyosis [9,11]. This is due to the fact that tamoxifen acts differently on receptors and has an oestrogenic effect in the myometrium and endometrium and an anti-oestrogenic effect in the breast, making it an important risk factor [24]. Postoperative analog GnRh can be used [26]. Vaginal radiotherapy ensures excellent local control [3].

The treatment is controversial [1]. Haemorrhage, intratumoural necrosis or rupture may be observed after surgery [7]. Stopping and reversing the epithelial–mesenchymal transformation could function as a therapeutic strategy [7]. Contrary to other clinical studies, this study suggests that the presence of adenomyosis is a favourable factor as it is a low-grade tumour with a better prognosis [11].

As rare pathologies, standard staging guidelines have not been established for cancers developed from adenomyosis [7]. As there is no staging, we use the pathology protocol advice [31]. Intramyometrial invasion is evaluated in terms of predicting regional lymph node metastasis [31].

The presence of two concurrent cancers changes the therapeutic approach, staging and prognosis [14].

## 10. Prognosis

According to some studies (Table 2), the prognosis is similar for endometrial cancer associated with adenomyosis and cancer that develops from adenomyosis, although the latter presents a more pronounced intramyometrial invasion, with muscle invasion being facilitated by adenomyosis [28].

The pathologist finds inflammatory reactions around adenomyosis in cases with a combination of endometrial cancer and adenomyosis. Other pathologies such as fibroids, polyps, and endometrial hyperplasia may also be associated with adenomyosis, all of which are influenced by hyperestrogenism [24].

The prognostic role of adenomyosis is still debated. Although the myometrial invasion is greater, the aggressiveness, according to some studies, is lower, and recurrences are rare [24]. The metastasis rate is similar to that of endometrial cancer not arising from adenomyosis. However, disease-free survival is lower, and patients are characterised as having a poor outcome [10].

The poor prognosis is due to invasion beyond the basal layer of the endometrium, with spread to the lymphatic and vascular system [10]. Therefore, lymphovascular space invasion (LVSI) is essential for prognosis [31,36]. LVSI influences therapeutic decisions, including external radiotherapy, especially if lymphadenectomy has not been performed [31].

The irregularity of the normal endometrial–myometrial junction makes it difficult to measure the depth of myometrial invasion. The overdiagnosis of superficial myometrial invasion may occur. Tumour involvement of adenomyosis may also result in problems assessing the depth of myometrial invasion. In this particular case, the tumour is not considered myoinvasive, and the prognosis is better [31]. Immunohistochemistry with CD10 can help determine prognosis by excluding a tumour developed from adenomyosis, improving the overall prognosis [2,31].

Endometrial cancer arising in adenomyosis occurs in significantly older patients [36]. Poor disease-free survival has also been observed, with the variant being more aggressive, but the overall survival is similar to that of cancer not arising from adenomyosis. However, the number of patients is too small in the literature to draw a clear conclusion [36]. Long-term follow-up is necessary to determine the prognosis [3]. Transvaginal ultrasound and CA125 are parts of standard follow-up practices, with MRI being an optional addition [3].

## 11. Discussions

It is important to explore genetics in these situations to understand the predisposition to malignant transformation of adenomyosis, as well as environmental factors, the role of oxidative stress and iron overload, lipid/protein and DNA damage, and associations with other pathologies, some with immunological and inflammatory components, such as Crohn’s disease, hepatocellular carcinoma and colon cancer. We need to understand the heterogeneity of this pathology for a more accurate approach to therapy [12].

Further studies are necessary, especially prospective ones, because thus far, the sample size has been small, with few cases to draw a conclusion [10]. The pathogenesis is still unclear [10].

The influence of adenomyosis and endometriosis in the development of cancer is unclear, but they seem to increase the risk of both endometrial and ovarian cancer [26]. The pathogenesis of adenomyosis and endometriosis is similar to that of other cancers, decreasing apoptosis and distant foci. Regarding malignant transformation, the results are contradictory [26]. However, in malignant transformation, hyperestrogenism, the immune system, inflammation, KRAS genes and the epithelial–mesenchymal transformation play crucial roles, sharing risk factors [26].

Regarding imaging diagnoses, there are no clear criteria for malignant transformation on MRI [26].

Consequently, there is no standardised management protocol or clear diagnostic criteria. It is also difficult to standardize [15]. There are no clinical treatment guidelines. Formal guidelines still need further research because an optimal treatment has not been established [3,21]. The efficacy of chemotherapy is also unknown [21]. Close and long-term follow-up is necessary.

The endometrium is a dynamic tissue with regenerative capacity under the influence of hormones that undergoes menstrual cycles. This fact is important in the pathogenesis of endometriosis, adenomyosis and even cancer [15]. The endometrium has two layers, functional and basal, with the latter being responsible for regeneration during the menstrual cycle [37]. These pathologies have common characteristics of endometriosis/adenomyosis/cancer and can even coexist, with common symptoms of chronic pelvic pain, abnormal bleeding and infertility [15].

The golden standard in diagnosis is a postoperative histopathological examination, and surgery is the golden standard in the treatment of these pathologies. There is currently no non-invasive diagnostic method, and the symptoms are nonspecific [15].

For almost a century, endometriosis and adenomyosis have been considered the same disease. The first systematic description belongs to Cullen, who described the invasion of the mucosa into the myometrium. Bird later provided the histological description [15]. Both endometriosis and adenomyosis present with an ectopic endometrium containing glandular epithelium and stromal components. Pathological findings of the endometrium depend on the duration of the disease and the degree of invasion, ranging from punctate foci and small stellate patches to cystic, nodular, or polypoid masses. The accumulation of hemosiderin, from unpigmented to red or blue dark lesions to white fibrotic scars or to chocolate cysts filled with blood may also be observed. Atypical presentations are possible. While an ASRM classification exists for adenomyosis (minimal, mild, moderate and severe), no such classification exists for endometriosis [15].

According to the literature, it is unclear how adenomyosis affects the occurrence and prognosis of uterine cancer, with the only consensus being that the malignancy rate is less than 1%. It has been reported that postmenopausal malignant transformation increases, probably due to cases of asymptomatic endometriosis and adenomyosis. Overtreatment with hormone replacement therapy has been avoided in recent years. Advances in oncology have had a significant impact with favourable outcomes, but follow-up data are still incomplete to provide accurate prognosis information [6].

According to the WHO, endometriosis is declared a potential precancerous lesion, and genetic involvement is significant. Common genetic alterations such as ARID1A, PTEN, p53 and KRAS are implicated in malignancy. The exact cellular pathways are unknown. Adenomyosis and endometriosis are closely linked [15].

Recently, a group of European and American researchers reviewed the literature on this topic and concluded that the lifetime risk of ovarian cancer among women with endometriosis is 1.8% compared to the general population. The incidence of ovarian cancer in cases of adenomyosis is not well evaluated, with fewer reported cases [14].

## 12. Future

The Wilms tumour 1 (WT1) marker is being considered a targeted treatment in cancer immunotherapy, particularly for its potential role in the immunoreactivity of uterine cancers [30]. The increased expression of MMP2 and MMP9 in malignancies is also being studied for the development of therapeutic concepts [18].

Next-generation sequencing (NGS) has shown that endometriosis and adenomyosis involve genetic alterations such as KRAS, PIK3CA, PPP2R1A and ARID1A. For therapeutic purposes, further research is needed in this direction to identify adenomyosis cases with a tendency for malignancy [16].

It has been established that there is a clearly defined clonal relationship between endometriosis and its malignant counterpart, underscoring the need to establish a similar relationship between adenomyosis and its malignancy [21].

IL-37b plays a role in various cancers, including liver and lung cancer, and appears to be important in gynaecological pathologies such as adenomyosis, endometriosis and cervical cancer. IL-37b is important in reducing the expression of matrix metalloproteinase 2 via the Rac1/NF-kB signalling pathway. Although it does not affect the epithelial–mesenchymal transition, IL-37b attenuates metastasis in endometrial cancer and provides a novel target in the diagnosis and treatment of gynaecological cancers [38].

There is a need for randomised controlled trials and extensive observational studies in this field.

## 13. Conclusions

Pathogenic mechanisms of transformation remain unclear, even if it has been established that adenomyosis, endometriosis and cancers may share a common pathogenic mechanism. Even in this context, malignancy is extremely rare.

As these cancers are very rare, it is difficult to standardize treatment. In this manuscript, we attempt to emphasize some aspects regarding management strategies and how heterogeneous the studies from the literature are in terms of prognosis in both cancers that develop from adenomyosis and those which are only associated with adenomyosis.

A novel target in the diagnosis and treatment of gynaecological malignancies associated with adenomyosis is also being studied for the development of new therapeutic concepts.

## Figures and Tables

**Figure 1 diagnostics-13-01883-f001:**
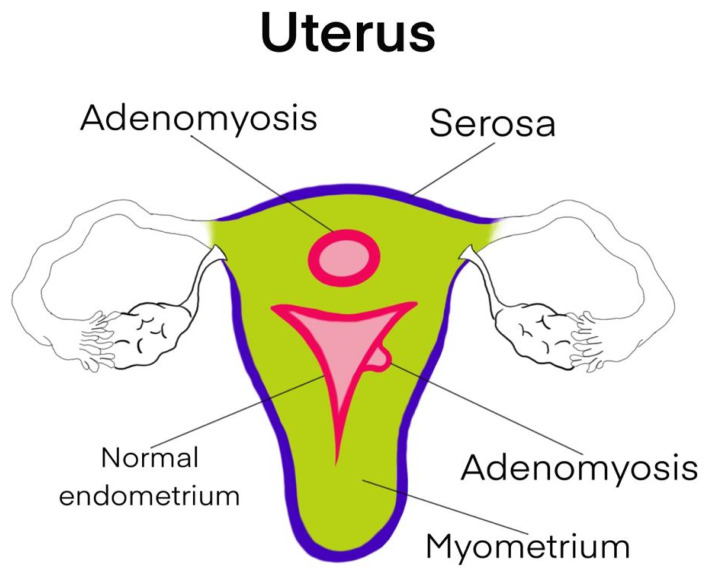
Schematic representation of adenomyosis.

**Table 1 diagnostics-13-01883-t001:** Common pathogenic mechanisms between adenomyosis, endometriosis and cancers.

Pathogenesis	Genetics	Inflamation and Immunology	Hormones	Oxidative Stress
Adenomyosis with a similar role in malignancy	K-RAS mutation (V-Ki-RAS 2 Kirsten rat) CTNNB1 encoding B-cathenin ARID1A (A-rich interactive domain-containing protein 1A) p53 JAZF1-SuZ12 EPC1-PHF1 PTEN loss	COX2 TNF-*α* Toll-like receptors (TLR1) Nuclear factor Kappa (NF-KB) Macrophage IL-6 IL-10	Oestrogen Poor response to progesteron	ROS Annexin ANXA2 EMT MMP2 and 9 (metalloproteinase)

COX2—cyclooxygenase-2; TNF-*α*—tumour necrosis factor alpha; IL-6—interleukin-6; IL-10—interleukin-10; ROS—oxygen-containing reactive species.

**Table 2 diagnostics-13-01883-t002:** Association of adenomyosis and endometriosis with cancer: prognosis and evolution.

Article	Number of Patients	Disease Free Survival (DSF)	Overall Survival	Prognosis	Mean Age (Range)	Muscle Invasion (≥½)	Influence of Adenomyosis on Cancer
Akiyo Taneichi [28] 2014 (A)	362 ▪ 121 (33.4%)—with adenomyosis; ▪ 241 (66.6%)—no adenomyosis;	No difference	No difference	Similar	Adenomyosis 56 years (32–84)	Significant *p* < 0.05 ▪ 19.5%—adenomyosis; ▪ 10.1%— non-adenomyosis;	No significant influence on prognosis.
Hiroko Machida [10] 2017 (B)	396 ▪ 46—EC-AIA (cancer arising from adenomyosis)—11.61%; ▪ 350—EC-A (cancer coexisting with adenomyosis)—88.39%	No difference	OS decreased in EC-AIA *p* = 0.031	EC-AIA—poor prognosis	EC-AIA 58.9 years	Significant *p* < 0.001 ▪ EC-AIA—51.6%; ▪ EC-A—19.4%	Unclear
Marjolein Hermens [26] 2021 (C)	129,872—enrolled ▪ 50,766 with endometriosis → 1827 developed endometrial cancer (3.59%); ▪ 85,051 with adenomyosis → 1408 developed endometrial cancer (1.65%)	No difference	No difference	Similar Adenomyosis has a poor response at hormonal treatment	Endometriosis 39 years (32–45)	Not investigated	The heterogeneity of the study is overdue, but endometriosis seems to have a greater influence than adenomyosis.
Koji Matsuo [36] 2015 (D)	1340 enrolled—all with endometrial cancer ▪ EC-AIA = 46 patients (3.43%); ▪ nonEC-AIA = 1294 patients (96.57%)	Significant poorer in EC-AIA *p* = 0.014	Significantly decreased in EC-AIA *p* = 0.001	Poorer in EC-AIA	EC-AIA 58.9 years (58.9 ± 9.9)	▪ EC-AIA—51.6%; ▪ nonEC-AIA—26.6%; Significant *p* = 0.002	With a lack of hormonal receptors in the cancer developed from adenomyosis ther was therefore a reduced hormonal response.

EC-AIA—endometrial cancer arising from adenomyosis; nonEC-AIA—endometrial cancer developed independently from adenomyosis.

## Data Availability

All data producedhere is available and can be produced upon request.

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
