# Peer review of "Adenomyosis and Its Possible Malignancy: A Review of the Literature"

_diagnostics, 2023, doi:10.3390/diagnostics13111883_

Round 1

Reviewer 1 Report

Thank you very much for the invitation to review of the manuscript. It a great pleasure for me.

The purpose of Moraru et al. was to emphasize some aspects regarding the adenomyosis and its possible malignancy. This is very interesting paper, and I have only one suggestion:

1.     In the Abstract section, it would be worth to add some conclusions.

2.     In the Introduction, there is no typical aim of the study.

3.     The searching strategy is very interesting, but very difficult to read. Could you do it as more readable version?

4.     Abbreviations should be explained

5.     Also, I am not sur if that part is about pathogenesis of adenomyosis or malignancy associated with the disease? Especially, that after you concentrate about diagnosis and symptoms of adenomyosis.

6.     What about other cytokines as cytokines from IL-1 family?

7.     What type of cells are involved in the process?

8.     it should be clearly specified which cancers are associated with adenomyosis

9.     Please avoid sentence equivalent. For instant point 5 of diagnosis:

„Associations with other types of cancer 344 Associations with other types of cancers include lymphoma, melanoma, ovarian, endometrial, or breast cancer. Synchronous cancers are more frequent, such as endometrioid adenocarcinoma of the ovary and clear cell carcinoma, and their coexistence modifies the approach [6,12,23].” I am not sure  what do you mean?

10.  Maybe it would be worth to move some information from “Future” part (about IL-37) to the introduction, as it is important in pathogenesis of the disease?

11.  The manuscript is a bit messed up. I would be worth to put all information with some order.

Author Response

Dear reviewer,

1). We added some conclusions to the abstract.[line 35-38]

2).The typical aim of our study, as you suggested is: Cancer arising from adenomyosis being very rare, adenomyosis, endometriosis and cancers may share a common pathogenic pathway, the pathogenic mechanism still remain to be established. Also there is a need to have a standardized treatment.[line 68-70]

3).The searching strategy is based on a PubMed data code related to our specific inclusion criteria written above[line 76-78].

4). All the abbreviations are explained.

5).We highlighted with track-changes the related parts to malignant transformation of adenomyosis in both, pathogenetic mechanism and diagnosis.

6,10). We included IL1 and moved a paragraph about IL37 to the section Pathogenesis.[ Line 161-166]

7). Both epithelial and mesenchymal histologic elements are involved in this highly malignant tumor[25].Histologically, glandular cells without atypia can be seen, along with sarcomatous components with pleomorphism and high mitotic rates [7].[line 316-320] We marked with track-changes parts in the histology part that underlines the types of cells involved in the malignancy process.

8).These types of cancers include lymphoma, melanoma, ovarian, endometrial, or breast cancer. Endometrial cancer can coexist with adenomyosis, without being developed from this [3,28]. Synchronous cancers are more frequent lately, such as endometrioid adenocarcinoma of the ovary and clear cell carcinoma, and their coexistence modifies the approach [6,12,23].[Line 376-380].

9).I rephrased and now I think is clearer what I intended to say:Other types of cancer can appear in association with adenomyosis without knowing any connection or predisposition in this regard.[line 375-376] Also this situation will change the therapeutical approach.[line436-437]

10) I moved from the Future section to Pathogenesis.

11).I tried to do as you say.

Thank you

Reviewer 2 Report

This manuscript is a review reporting adenomyosis and its possible malignancy. The author also reviewed the management strategy. This was a comprehensive literature review article, and the authors have collected a unique dataset. This paper was methodologically quite sound and generally well-written. I believe the paper might be acceptable for publication following a minor revision of the points below.

1.     The introduction and the methodology are too brief for this high-impact journal. I suggest the authors add more content to these sections.

2.     Please explain the abbreviation in a footnote in Table 2.

Minor editing of English language required.

Author Response

Dear reviewer

1).We added more relevant content to introduction and methodology.And in addition we emphasized some important points in the text.[59-70,80,85]

2) I explained the abbreviations in table 2 in the footnote.[line 475]

Thank you

Round 2

Reviewer 1 Report

The purpose of Moraru et al. was to emphasize some aspects regarding the adenomyosis and its possible malignancy. This is very interesting paper, and I have only one suggestion:

1.     Still, the searching strategy is very difficult to read. Could you do it as more readable version?

2.     Again, the manuscript is a bit disorganised. Some information should have been in different order.

Author Response

Cover letter

Dear reviewer

1).I emphasize the way we used the PubMed search method in a more readable and understandable way.

2) I changed the order, I hope it is better now.

Thank you